# Conserved Control Path in Multilayer Networks

**DOI:** 10.3390/e24070979

**Published:** 2022-07-15

**Authors:** Bingbo Wang, Xiujuan Ma, Cunchi Wang, Mingjie Zhang, Qianhua Gong, Lin Gao

**Affiliations:** School of Computer Science and Technology, Xidian University, Xi’an 710071, China; xjuanma@stu.xidian.edu.cn (X.M.); wangcunchi@stu.xidian.edu.cn (C.W.); mjzhang720@stu.xidian.edu.cn (M.Z.); violetgong@163.com (Q.G.)

**Keywords:** multilayer network, control path, pan-cancer, controllability

## Abstract

The determination of directed control paths in complex networks is important because control paths indicate the structure of the propagation of control signals through edges. A challenging problem is to identify them in complex networked systems characterized by different types of interactions that form multilayer networks. In this study, we describe a graph pattern called the conserved control path, which allows us to model a common control structure among different types of relations. We present a practical conserved control path detection method (CoPath), which is based on a maximum-weighted matching, to determine the paths that play the most consistent roles in controlling signal transmission in multilayer networks. As a pragmatic application, we demonstrate that the control paths detected in a multilayered pan-cancer network are statistically more consistent. Additionally, they lead to the effective identification of drug targets, thereby demonstrating their power in predicting key pathways that influence multiple cancers.

## 1. Introduction

Understanding the control properties of complex systems provides us with insight into how designers interfere with these systems to achieve desired behaviors and helps us to understand the dynamic process [1]. In the past few decades, many approaches have been developed to study the controllability of single-layer networks [2], such as control based on maximum matching [3], the minimum dominating set [4,5], and the feedback vertex set [6]. Liu et al. identified the minimum number of driver nodes required to maintain full control of a network by solving a maximum matching problem, which provides an efficient method to determine the driver nodes for arbitrary directed networks and provides a type of classification for edges [3].

Moreover, for more complex systems, they can be better modeled by multilayer network structures than single-layer networks to help us to understand the system more fully. For instance, in social networks, multilayer networks created according to different relationships are clearer than single-layer networks [7,8]. There have been increasingly intense efforts to investigate networks with multiple types of connections. A social network can be represented by a set of people (nodes) with some patterns of contacts or relations (links) between them. Multiple layers indicate multiple relationships, including colleagueship, friendship, partnership, and membership of the same community [9]. Each layer has a specific topological pattern that can spread information between the current layer and other layers if links exist among different layers. In multilayer biological networks, Zheng et al. [10] confirmed that, in the minimum driver nodes they identified, some nodes can act as drug targets in biological experiments and other nodes are also involved in important biological processes. Compared with the minimum set of driver nodes identified in single-layer networks, more drug targets occurred in a multilayer network [11,12,13]. Liu et al. [14] demonstrated that remote control can enhance the robustness of multilayer transportation systems against cascading failure. Simultaneously, the controllability of multilayer networks has become increasingly important, and researchers have paid much attention to it. Wang et al. [15] explored the effects of different interlayer couplings on the controllability and control energy cost of multilayer networks. Yuan et al. [16] analyzed the controllability of multilayer networks based on maximum multiplicity theory. Nacher et al. [17] developed FAST-MDSM to identify the minimum set of driver nodes required to control multilayer networks.

However, understanding the control properties of a system requires not only knowing how many drivers are required but also characterizing the propagation pattern of control signals. Directed control paths, called stems, illustrate the backbone structure of the propagation of control signals through the network. Importing one signal for each control path guarantees controllability. These paths then transmit signals to control the entire system, including cycles that are inherently self-regulatory [18]. The control paths characterize the functional origin of each control and provide insight into the high-level organizations and function of complex systems. In previous studies, researchers showed the effective outcome of directed control paths and controllability in directed networks, particularly biological networks [19,20].

Despite advances in controlling multilayer networks, to date, a deep understanding of the properties of control paths in multilayer networks has not been conveyed in the literature. In this paper, we propose a pattern, index, and algorithm to study fundamental issues pertinent to the control paths of multilayer networks and obtain some interesting results. We define a novel graph pattern, that is, conserved control paths (CCPs), to model the common elements of control paths in different layers, which allows us to identify a consistent control structure among different types of relations. Additionally, we propose the conservative degree (CV) index as a metric to quantify the consistency of control paths. We present a practical CCP detection algorithm (CoPath) to detect CCPs from *L*-layer networks. CoPath can be implemented in low-order polynomial time $O\left(L\times n^{3}\right)$. We used the index CV to test the consistency of control paths in synthetic scale-free (SF) and Erdos–Rényi (ER) networks. The results demonstrated that the consistency of control paths changed according to the number of layers and network density, but it was robust under the disturbed weight of edges. Furthermore, we applied CoPath in pan-cancer analysis. We constructed a 16-layer pan-cancer network and detected its CCPs among 16 types of typical cancers. This pan-cancer network had more consistent control paths among layers than its randomized counterparts. The CV reduced by 58.2% when we disturbed each layer by crossing only 8% of edges. We demonstrated that CCPs among cancers tended to be conserved, and there were potential common graph structures between different cancers. In particular, we found that drug targets tended to be located upstream of these CCPs (*p*-value = 0.0062), which verified the vital role of the CCP pattern in delivering drug effects to influence cancers. The CCP pattern, CV index, and CoPath algorithm provide a reasonable framework to help us understand control properties and are practical in the functional pathway detection of multilayer networks.

## 2. Materials and Methods

### 2.1. Control Path Based on Structural Controllability

A linear time-invariant dynamic system x˙(t)=Ax(t)+Bu(t) is said to be structurally controllable if and only if the n × nm controllability matrix C≡[B, AB, A2B, …, An−1B] has full rank; that is, if rank(C)=n, the system can be driven from any initial state with appropriate input control signals to any desirable states within a finite time [3,21]. The state vector x∈ℝn captures the state of all *n* nodes at time *t*. A∈ℝn×ℝn is the adjacency matrix of the underlying directed network of this system, B∈ℝn×ℝm is the input matrix, u∈ℝm is the input vector, *m* is the number of driver nodes, and *n* is the number of nodes. External signals u(t)={u1(t), …, um(t)}′ are imposed on the *m* drivers (receive external signals, m ≤ n). For example, in Figure 1a panel (ii), u(t)={u1(t), u2(t)}′ indicates that control signals u1 and u2 are imposed on node 1 and node 4, respectively.

For structural controllability, Liu et al. [3] innovatively proposed the maximum matching set (MMSet, denoted by *M*) to assess and quantify the size of the minimum driver node set (MDSet) required to fully control an arbitrary network. They indicated that the set of unmatched nodes in *M* is exactly MDSet: |MDset|=max{n−|M|, 1}. M is matched subject to the following: (i) M⊆E is an edge set with maximum cardinality, and (ii) no two edges in M share a starting node or ending node. In Figure 1a, the given network contains five nodes and the MMSet has three matching edges. A potential MMSet is highlighted in purple in panel (ii). The two unmatched nodes (max{(5−3), 1}, node 1 and 4) that are not pointed to by purple edges are drivers.

Moreover, edge set M forms a stem-cycle cover of the original network, and starting from the unmatched drivers, the input signals are transmitted along the directed edges in *M* to guide the entire network to any desired final state [18]. Therefore, Wang et al. [22,23] proposed that the MMSet not only reveals the MDSet but also consists of the backbone of the key control paths. Edges in M are directed control flows from the driver nodes to all matched nodes. These directed control flows are called control paths, which are related to the dynamical process of propagating the perturbation influence from drivers. For example, in Figure 1a panel (ii), the purple matching edges of the MMSet form two directed control paths 1→2→3 and 4→5. By injecting control signals into the roots (node 1 and node 4) of these two paths, the control signal is transmitted to other matched nodes along the matching edges to fully control this monoplex system. As another example, in Figure 1b, control paths corresponding to the purple MMSet in the first layer are 1→2→3→6 and 4→5→7.

However, there are diversified MMSets (n! matchings for a complete connected network that has *n* nodes). Each MMSet illustrates a unique set of control paths through which control influences are transmitted under the minimum cost condition, and no exact polynomial algorithm can determine the number of M in an arbitrary network. Counting the number of M was proven to belong to the #P-complete class of problems [24]. Therefore, several methods count the maximum matchings approximately for some special network types [24,25]. Zdeborová et al. [24] found that the number of maximum matchings increases exponentially as the mean degree grows in ER random graphs. Liu et al. [25] provided two lower bounds on the number of maximum matchings of bipartite graphs with a positive surplus. The lower bounds are related to the number of nodes and edges. In such a scenario, control paths in a directed graph are not unique and the number of maximum matchings for large-scale networks is not small. Therefore, researchers have attempted to use alternative solutions to investigate large-scale networks that require information about all maximum matchings. Jia et al. [26] proposed a random sampling method to explore the controllability of complex networks instead of enumerating all MMSets for a monoplex network. We show two different MMSets of the network in Figure 1a panels (ii,iii). Different from panel ii, the network can also be fully controlled by imposed signals on two additional driver nodes 1 and 3, and then one directed path 1→2→4→5 is identified as the control route.

Furthermore, based on the role of edges in diversified MMSets, edges *E* can be classified into three categories: “critical” edges $E^{c}$ must appear in all MMSets (i.e., the absence of “critical” edges increases the size of an MDSet); “redundant” edges $E^{r}$ may never appear in any MMSets (i.e., the absence of “redundant” edges decreases the size of an MDSet); and “ordinary” edges $E^{o}$ have roles in some MMSets (i.e., the absence of “redundant” edges does not affect the size of an MDSet) [3]. Different MMSets have the same critical edges but variant ordinary edges to make them different. Additionally, there are always many “redundant” edges that have no association with any control routes, and the existence of “critical” edges indicates the changeless part of all control routes. The categories of links can be determined by the algorithm proposed by Régin [27]. In Figure 1a, purple edges in panels ii and iii form two MMSets for the network in panel i. By definition, 1→2 and 4→5 are critical edges because they appear in both MMSets; 2→5 is a redundant edge because it never appears in either MMSet; and 2→3 and 2→4 are ordinary edges because they occasionally appear in the MMSets.

### 2.2. Conserved Control Path

To investigate the common control pattern of a system with multiple types of connections, we use a multilayer network Γ that contains *L* layers and can be described by $\Gamma =\left(\left\{G_{1},\;G_{2},\;\ldots ,\;G_{L}\right\},\;\Phi \right)$. $G_{l}=\left(V_{l},\;E_{l}\right)$ represents the *l*-th layer in Γ, and the crossed layer interconnections between any two layers $G_{l}$ and $G_{k}$ are denoted by ∅. We describe a graph pattern, that is, CCPs, that allows us to model common control structures among different types of relations of Γ.

**Definition** **1.***For a given maximum matching $M_{l}$ of a monoplex network $G_{l}$, a control path cp is defined to be the largest connected sequence of nodes $cp=\left\{(i_{1},\;i_{2}),\;\left(i_{2},\;i_{3}\right),\;\ldots ,\;\left(i_{K-1},\;i_{K}\right)\right\}$, $i_{k}\in V_{l}$, 1≤k≤K subject to $\forall k,\;\left(i_{k-1},\;i_{k}\right)\in M_{l}$*.

$M_{l}$ forms a stem-cycle cover of $G_{l}$, where each stem or cycle is the largest connected sequence of nodes, which is defined as a control path *cp*. Additionally, the number of stems equals the minimum number of driver nodes required to fully control $G_{l}$, by injecting signals into the roots of stems. Then the signals are transmitted to the entire network through control paths. As an example, in Figure 1b, the control paths shown in purple that correspond to an MMSet in the third layer are 2→3→ 6 and 4→5→7→8.

**Definition** **2.***For a given set of maximum matchings $ℳ=\left\{M_{1},\;M_{2},\;\ldots ,\;M_{L}\right\}$ of multilayer network $\Gamma =\left(\left\{G_{1},\;G_{2},\;\ldots ,\;G_{L}\right\},\;\Phi \right)$, $M_{l}$ is a maximum matching of $G_{l}$ and the CV of the control paths is defined as $CV\left(\chi \right)=\left(1/\left|\chi \right|\right)\sum_{e\in \chi }{\left(\left(1/L\right)\sum_{l=1}^{L}I_{e\in M_{l}}\left(e\right)\right)}^{2}$, $\chi =\underset{l}{\cup }M_{l}$, 1≤l≤L*.

χ is a union set of maximum matchings of every layer. I() is an indicator function, where I(e)=1 when $e\in M_{l}$; otherwise, I(e)=0. CV(χ) quantifies the CV of an edge set χ. The maximum CV(χ)=1 when ∀i, j (1≤i<j≤L) and $M_{i}=M_{j}$, indicates that all layers can be controlled by identical control paths. The minimum $CV\left(\chi \right)={\left(1/L\right)}^{2}$ when ∀i, j (1≤i<j≤L) and $M_{i}\cap M_{j}\;=\;\emptyset$, that is, there are no common control paths between any pair of layers, indicates exceedingly different control patterns between layers. In Figure 1b, *L* = 3, and there are two control paths corresponding to the maximum matching: 2→3 is one edge that is present in the control paths of all three layers and $\sum_{l\;=\;1}^{3}I_{(2\to 3)\in M_{l}}\left(2\to 3\right)\;=\;3$. For these two CCPs, 1→2→3→6 and 4→5→7→8, χ={(1, 2), (2, 3), (3, 6), (4, 5), (5, 7), (5, 8)}, and the final CV(χ)= 0.67.

**Problem** **1.*****Conserved control path detection.** Given a multilayer network $\Gamma =\left(\left\{G_{1},\;G_{2},\;\ldots ,\;G_{L}\right\},\Phi \right)$, determine a corresponding $\hat{ℳ}=\left\{M_{1},\;M_{2},\;\ldots ,\;M_{L}\right\}$ with the maximum, C.V.; $\hat{ℳ}=\underset{ℳ}{\arg\max}CV\left(\underset{l}{\cup }M_{l}\right)$*.

However, how to detect $\hat{ℳ}$ efficiently is an opening issue. A direct approach is to solve all MMSets of every layer ℳ in Γ and find the best combination of maximum matchings in all layers ℳ that has the most conserved edges; however, enumerating all maximum matchings of a large graph is impractical. Additionally, enumerating all combinations of maximum matchings for all layers requires a vast amount of calculation. Suppose we consider a 10-layer network and there are only 10 maximum matchings for each layer. At most, there are $10^{10}$ combinations for the 10-layer network. The combinatorial explosion of enumerating all MMSets is a challenge for large-scale multilayer networks with thousands of maximum matchings when we only want to identify a specific combination of different MMSets in a multilayer network. To circumvent such an approach because of its high computational complexity, we develop a more effective algorithm to determine an approximate solution of $\hat{ℳ}$.

Because only ordinary edges are optional in different maximum matchings, we can only detect the conserved ordinary edges in Γ, and then add the necessary critical edges to restore the control path. Redundant edges are futile in the control path.

**Definition** **3.***For a monoplex network $G_{l}=\left(V_{l},\;E_{l}\right)$, its ordinary induced sub-network (OIN) is $G_{l}^{o}=\left(V_{l}{}^{o},\;E_{{}_{l}}^{o}\right)$, where $E_{{}_{l}}^{o}$ is a set of ordinary edges*.

As shown in Figure 1c, the ordinary edges in the first layer are 1^+^→2^−^, 1^+^→3^−^, 2^+^→3^−^, 2^+^→4^−^, and 4^+^→5^−^, and they constitute its OIN. The OIN is presented as a bipartite graph with add and subtract signs on the node marks (details are in the following section).

**Problem** **2.*****Conserved control path detection in OINs.** Given an L-layer network $\Gamma =\left(\left\{G_{1},\;G_{2},\;\ldots ,\;G_{L}\right\},\Phi \right)$, determine ${\hat{ℳ}}^{o}=\left\{M_{1}^{o},\;M_{2}^{o},\;\ldots ,\;M_{L}^{o}\right\}$ with the maximum CV $\hat{ℳ}=\underset{ℳ}{\arg\max}CV\left(\underset{l}{\cup }M_{l}^{o}\cup E_{l}^{c}\right)$ in L-layer OINs $\Gamma^{o}=\left(\left\{G_{1}^{o},\;G_{2}^{o},\;\ldots ,\;G_{L}^{o}\right\},\;\Phi \right)$, where $M_{l}^{o}$ is the maximum matching of OIN $G_{l}^{o}$ and $E_{l}^{c}$ is the set of critical edges of $G_{l}$*.

Based on the category of edges, we can remove redundant edges and determine critical edges in advance. Then we focus on the ordinary edges, which allows the algorithm to be sufficiently optimized to further reduce the running time and improve efficiency. The time complexity of an algorithm for detecting the CCP is $O\left(n^{3}\right)$, where *n* is the number of nodes in the network.

Despite the fact that we can achieve such improvements in advance, it remains difficult to enumerate all MMSets of every layer in $\Gamma^{o}$ to determine the optimal solution $\hat{ℳ}$ because of the combination explosion of MMSets in all layers for a large-scale multilayer network. Therefore, we provide an efficient approach to obtain an approximate solution of $\hat{ℳ}$. We convert the problem of solving a set of CCPs in a multilayer network into solving the CCPs of each layer $G_{l}^{o}$ in the network to identify an optimal solution $M_{l}^{o}$ in monoplex $G_{l}^{o}$ of $\Gamma^{o}=\left(\left\{G_{1}^{o},\;G_{2}^{o},\;\ldots ,\;G_{L}^{o}\right\},\;\Phi \right)$. The optimal solution $M_{l}^{o}$ is a set of the most conserved CCPs with the maximum conservative degree in $G_{l}^{o}$. Then we combine $M_{l}^{o}$ of each layer $\underset{1\leq l\leq L}{\cup }M_{l}^{o}\cup E_{l}^{c}$, which is an approximate solution of $\hat{ℳ}$.

**Definition** **4.***For a given multilayer network $\Gamma =\left(\left\{G_{1},\;G_{2},\;\ldots ,\;G_{L}\right\},\;\Phi \right)$, $E_{l}^{o}$ and $E_{l}^{c}$ are sets of ordinary and critical edges in $G_{l}$, respectively. The conservative weight of an edge e is defined as $\hat{w}\left(e\right)=\sum_{l=1}^{L}I_{e\in \left(E_{l}^{o}\cup E_{l}^{c}\right)}\left(e\right)$*.

The index conservative weight is described as the number of layers in which an edge appears in a multilayer network composed of ordinary edges and critical edges. It represents the conservative probability of edge *e* in $\Gamma^{o}$. Then, with a given weighted $\Gamma^{o}$, we can convert the problem of enumerating all MMSets to obtaining the most conserved (maximum weight) control path set of $G_{l}^{o}$. In Figure 1c, $\Gamma^{o}=\left(\left\{G_{1}^{o},\;G_{2}^{o},\;G_{3}^{o}\right\},\;\Phi \right)$, and the conservative weights of some ordinary edges in $G_{1}^{o}$ are $\hat{w}$(1^+^→2^−^) = 2, $\hat{w}$(2^+^→3^−^) = 3 and $\hat{w}$(4^+^→5^−^) = 3. The thickness of the line is directly proportional to the conservative weight of the edges. The conserved ${\hat{ℳ}}^{o}=\left\{M_{1}^{o},\;M_{2}^{o},\;M_{3}^{o}\right\}$ with the maximum CV can be identified as the maximum-cardinality matching with maximum weight as described in the following section. For $G_{1}^{o}$, the most conserved (maximum weight) control paths $M_{1}^{o}$ are {(1, 2), (2, 3), (4, 5)}.

### 2.3. Conserved Control Path Detection Method

To solve the above problems, we propose a novel process called CoPath (see Algorithm 1) to detect ℳ^ efficiently. For a given multilayer network, conservative degree of an edge is taken as the respective weight. Through detecting the maximum-cardinality matching with maximum weight among all maximum-cardinality matchings to obtain conserved control paths. Maximum-cardinality matching ensures matching edges for the corresponding control path are obtained and the maximum weight is for maximum conservatism between layers. Simultaneously, referring to the classification of edges in controllability, we remove redundant edges in our process for their irrelevant role in the control process, just to further improve efficiency of CoPath. First, referring to the classification of edges by Liu et al. [3], we classify edges for each layer in a multilayer network into critical edges, ordinary edges, and redundant edges, respectively. Second, for each Glo∈Γo, we construct a new weight w^(e) for every edge in Glo and refer to Γo with w^(e) as weighted Γo. According to the definition of edge classification, we do not consider redundant edges in the process of detecting ℳ^ with the maximum CV. Because redundant edges do not play a vital role in the control process, removing the many redundant edges improves efficiency. Third, we identify the most common edge set ℳ^o={M1o, M2o, …, MLo} with the maximum CV using the Python tool networkx.algorithms.matching.max_weight_matching [28]. Finally, we add critical edges to restore the control path.
**Algorithm 1**. CoPath(1) **Input:** A directed multilayer network Γ=({G1, G2, …, GL}, Φ).(2) **Loop:** For each layer *l* ∀Gl∈Γ;   1: Classify *E_l_* into (Elc, Elr, Elo);   2: Assign weight w^(e) in Glo=(Vlo, Elo);   3: Construct a bipartite network BGlo(V+, V−, E±);   4: Mlo = maximum-cardinality matching with maximum weight (BGlo);   5: Ml← add Elc to Mlo.(3) **Output:**ℳ^={M1, M2, …, ML}

The schematic diagram of CCPs detected by CoPath for a three-layer network (layer *l* = 1, 2, 3) is shown in Figure 1c. First, we determine the critical edges $E_{l}^{c}$ shown in pink in advance and remove redundant edges $E_{l}^{r}$ shown as black dashed lines for each layer. Because critical edges must occur in all MMSets for a network and redundant edges must never occur in any MMSets, we do not need to consider different combinations of critical edges and redundant edges when detecting CCPs, but only consider the ordinary edges $E_{l}^{o}$ shown as black solid lines. Second, we construct ordinary induced sub-networks for the three-layer network, and the thickness of solid lines reflects the conservative weight $\hat{w}\left(e\right)$ of ordinary edges. Third, we convert the ordinary induced sub-networks to bipartite graphs $BG_{l}^{o}$. For example, edge 2→3 in the original network in Figure 1b corresponds to edge 2^+^→3^−^ in Figure 1c. Fourth, we detect the maximum-cardinality matching ${\hat{ℳ}}^{o}=\left\{M_{1}^{o},\;M_{2}^{o},\;M_{3}^{o}\right\}$ with maximum weight for a bipartite graph of each layer. For the three-layer network, the result is {(1, 2), (2, 3), (4, 5)} for the first layer, {(1, 2), (2, 3), (4, 5)} for the second layer, and {(3, 6), (5, 7)} for the third layer. We detect edges with larger weights for each layer. Finally, we add the essential critical pink edges to restore the control path in each layer $\hat{ℳ}=\left\{M_{1},\;M_{2},\;M_{3}\right\}$. CCPs CCP1 and CCP2 are shown on the right of Figure 1b.

CoPath has the following advantages: (a) we only consider ordinary edges when calculating maximum-cardinality matching with maximum weight, which shortens the running time, and the complexity of our proposed algorithm is $O\left(n^{3}\right)$; (b) each layer can be an unweighted or weighted network; and (c) we do not require the same nodes in every layer.

## 3. Results

### 3.1. Conserved Control Path Pattern in Synthetic Networks

To evaluate which factors influence the CCP pattern in a complex system and how changes to the parameterization of multilayer networks affect the, C.V.; we conducted experiments on SF network [29] and ER network [30]. As both types of networks achieve similar results, we only discussed the results of the SF network (Figure 2). The generation of the SF network refers to the study by Jia et al. [26]. Moreover, we extended their method to construct a multilayer network (see Appendix A).

To test the effects of the number of layers and the proportion of common edges (see Appendix A) in different layers on the, C.V.; we detected the CCPs and quantified the consistency of the CCPs across all layers by the index CV. We found that the CV decreased as the layers increased and common edges decreased among different layers (Figure 2a). For example, when the proportion of common edges was 0.9 between any two layers, the CV decreased from approximately 0.5 to 0.14 as the number of layers increased in a 10-layer network. However, comparatively, the CV of a completely random network was approximately 0.01, which explained that the CCPs occurred if public mechanisms existed in different layers of the multilayer network. To test the robustness of the CCPs, we added four types of Gaussian noise (μ=0, σ=1, 3, 5, 25) to the weight of the edge (see Appendix A) in each network (the same network that we used in Figure 2a) and compared the CVs before and after adding noise. We noticed that there was no obvious difference in the CVs before and after (dashed line) adding the four types of noise (Figure 2b). Furthermore, we measured the ratio of overlapping edges in the CCPs before and after adding noise, which demonstrated that noise had only a small influence on the CCPs when there was a specified proportion of common edges among different layers (Figure 2c).

In the same manner, we found that the CV decreased gently as the different nodes between adjacent layers increased (Figure 2d), see Appendix A), which indicated that CoPath mainly detected the overlap of edges in different layers, and it was slightly affected by the differences in a small number of nodes. Moreover, we observed that the network density had a big influence on the CV (Figure 2e). We believed that this phenomenon can be explained by the proportion of ordinary edges varying with the average degree 〈k〉 of the network. Referring to the results of Liu et al. [3], the number of ordinary edges increased when 〈k〉 of the ER network ranged from 0 to 2 and it varied little from 2 to 2e (e is a natural constant, approximately 2.71828). When 〈k〉 was 2e, the number of redundant edges was the highest, and the number of ordinary edges increased again as 〈k〉 increased. From Figure 2e, there was also an interesting phase change. The CV decreased when the number of ordinary edges increased, which corresponded to a network density ranged from 0.0025 to 0.005, and the CV demonstrated a rising tendency, which corresponded to a network density range from 0.005 to 0.015. The obvious phase change was that the CV decreased when the network density was larger than 0.015. We believed that the phase change occurred because the proportion of common edges was set to be constant in our synthetic networks. We preserved 50% of the common edges from the layer with the fewest nodes across different layers regardless of the network density. Therefore, when the network density increased from 0.005 to 0.015, increasingly common ordinary edges between layers existed. However, after 0.015, when we achieved 50% of the common edges, there was only an opportunity to import specific ordinary edges into each layer, and this led to a decreasing CV at the phase change.

Considering these results together, we found that the CV was indeed affected by some factors, such as the number of layers and proportion of common edges in different layers. However, it was minimally affected by the original weight of the edges, which demonstrated that CoPath focused prominently on the network structure.

### 3.2. Application in Pan-Cancer

Different cancers are treated separately, and substantial differences are presumed to exist among most cancers. Despite this variability, some cancers demonstrate some similarities, for example, COAD and READ have similar patterns of genomic alteration [31], which prompts us to study common structures among different cancers. Additionally, a pathway is considered to be the smallest functional unit of a network that can perform a single task. The development of cancer is related to the dysregulation of related biological pathways [32]. From the view of control theory, driver nodes exist that play a vital role in cancers receiving disease-promoting signals, and thus influence other nodes in the network. Hence, we applied CoPath, which was based on control theory, to a pan-cancer network. We expected to determine some CCP patterns for different cancers and provide a new direction for further research. The framework of pan-cancer application, including: (1) Collect the human signaling network as the background network from pathway commons database. (2) Quantify activity of edge which is perturbed by a cancer based on The Cancer Genome Atlas (TCGA) datasets and construct a cancer-specific activated signaling network. (3) Detect CCPs in 16-layer cancer-related activated signaling networks. (4) Investigate biological functions of CCPs based on enrichment analyses in functional gene sets and pathways. (5) Validate the key roles of CCPs in drug target prediction (F1-measure) and disease treatment mechanism identification. We make efforts as follows to solve these issues in turn.

We first collected 16 common cancer-related data and constructed a 16-layer cancer-related activated signaling network (see Appendix A). The details of the scale and other information of the 16-layer network based on edge classification are listed in Table 1.

Hereafter, we associated the CCPs for the 16-layer network of cancers with CoPath (Appendix A). To test whether the real cancer networks had higher consistency than their randomized counterparts, we compared the CV of the CCPs in the real network with that of the CCPs in the perturbed network generated by randomly selecting two edges from the real network and crossing them, which did not change the number of edges and network density of the original network. We found that the CV decreased sharply as more edges were crossed (Figure 2f), and the CV of the 16-layer cancer network (0.165) was much higher than the values of its randomized counterpart networks (0.069 with 8% perturbed edges). This indicated that the CCPs among cancers were inclined to be conserved and there were potential common graph structures for cancers; that is, cancer was conservative in its mechanism of action, which further prompted us to analyze the CCPs of a multilayer cancer-related activated signaling network.

We assessed the CV of edges in the CCPs (Figure 3a), which demonstrated that the path we detected was indeed conserved, with only one exception of ESCA. We considered that the obvious difference in the number of nodes and edges between ESCA and other cancers may be the main reason for this disparate result. Although there was a very weak correlation between the conservative weight and the activity of edges in the cancer-related activated signaling network (Pearson correlation coefficients for 16 cancers were in the range of −0.04 to 0.25), generally, the activity of ordinary edges in the CCPs was higher than those not in the CCPs, except for ESCA.

To investigate whether the genes in the CCPs had significant biological importance, we performed two enrichment analyses to access them using the hypergeometric distribution. We considered nine functional gene sets (Appendix A) and listed the number and sources in Table 2.

We present all the results of the enrichment analyses for the nine functional gene sets in Appendix A. In this section, we present the details of the enrichment analysis for drug target genes. First, we assessed whether the genes in the CCPs or elsewhere were enriched for the functional gene sets. We considered genes in the network as nodes and divided them into isolated (nodes not in the CCPs), path (nodes in the path of the CCPs), and cycle (nodes in circular CCPs) nodes according to their positions in the networks. We found that genes in the paths of the 16 cancers we studied were significantly enriched for the nine functional gene sets, except genome-wide association study (GWAS) disease genes in ESCA and READ (Figure 3b). Second, we divided the nodes in the paths of the CCPs into source, middle and sink based on their positions. We observed that drug target genes were significantly enriched in the middle position of the CCPs (Figure 3c). Additionally, similar results existed for other functional gene sets, except virus host genes, for which the source nodes were enriched. Based on the results shown in Figure 3c, we considered whether functional genes were located upstream or downstream of the CCPs. This information would allow us to measure the number of downstream genes in the CCPs for functional genes and non-functional genes. We observed that drug target genes had more downstream genes than non-drug target genes in the CCPs, which indicated that drug target genes may occupy locations upstream of the CCPs (Figure 3d). To verify this, we performed a further analysis, which was a further verification for Figure 3d; that is, the weight of the downstream path of one node in the CCPs, which we calculated by summing the conservative values ($\sum_{l=1}^{L}I_{e\in M_{l}}\left(e\right)$) of edges composed of the node and its downstream genes. Once again, this implied that the conservative weight of the downstream path of drug target genes was larger than that of non-drug target genes (Figure 3e). Moreover, we accessed the statistical significance of drug target genes located upstream of the CCPs. We emphasized the significant differences (87.5% cases with *p*-values < 0.05 in (d) and 75% cases with *p*-values < 0.05 in (e)) using the rank-sum test between drug target genes and non-drug target genes. We found that the difference between two cancers (ESCA and READ) was not significant in the analysis of Figure 3d. Additionally, the difference among four cancers (ESCA, KICH, LIHC, and READ) was not significant in the analysis of Figure 3e. These results explained the importance of cancer-related CCPs and indicated that CCPs were helpful for identifying drug targets.

To investigate the function of CCPs, we analyzed pathway-based enrichment for BLCA and BRCA as examples. There were 1165 CCPs (containing 3371 genes) and 1246 CCPs (contained 3743 genes) for BLCA and BRCA, respectively. For each cancer, we considered every *cp* in the CCPs and took the genes of the *cp* as the input list to analyze functional pathways for which *cp* could be enriched (see Appendix A). We obtained 520 functional pathways for BLCA and 621 functional pathways for BRCA. There were 390 common pathways for BLCA and BRCA, which demonstrated that pathways enriched by the CCPs were consistent in these two cancers. Additionally, we manually checked whether pathways, which were enriched by the CCPs, were related to cancer. Because a *cp* containing fewer than six genes was enriched for fewer functional pathways, we examined a long *cp* containing at least six genes. We found that at least 87.34% (69/79) of the CCPs in BLCA (85.85%, 85/99 of the CCPs in BRCA) were enriched for cancer-related pathways. In Table 3, we listed ten pathways that occurred frequently in both cancers (i.e., many different CCPs were enriched for these 10 pathways). The pathway-based enrichment of BLCA and BRCA demonstrated that the two cancers had some common mechanisms, which explained the conservatism of the CCPs detected by CoPath. By contrast, some enriched pathways of the CCPs were related to cancers that were demonstrated in previous studies. For example, the most enriched pathway was the metabolism pathway contributing to oncogenic mutations [34]. Generally, cancer cells exhibited increased lipogenesis, which has been proposed as a target for cancer treatment, and suggested the role of the metabolism of the lipid pathway in cancers [35]. It further indicated that the CCPs of a cancer-related activated signaling network were valid and of biological importance.

Then, to further illustrate the reliability of the CCPs of the 16-layer cancer network, we measured the similarities between cancers based on the Jaccard index of the CCPs. We questioned whether the similarities between cancers calculated using CCPs (termed CCP-based similarity) were correlated with the similarities in previous studies. We compared CCP-based similarity with a type of well-studied disease similarity [36] based on Medical Subject Headings (MeSH). We studied 11 out of 16 cancers and obtained the similarities between them. Then, we fitted a straight line for CCP-based similarity and MeSH similarity in Figure 4a, which showed that two similarity measures between cancers have a certain positive correlation. As an example, we found a cancer pair BRCA and COAD that had higher CCP-based similarity than the MeSH-based similarity (Figure 4a), which further reflected the conservative characteristics of the CCPs. Furthermore, we found that approximately 90.06% (3371 of 3743) genes of BRCA and 92.28% (3371 of 3653) genes of COAD in the CCPs overlapped. Approximately 63.14% (1598 of 2531) edges of BRCA and 64.78% (1598 of 2467) edges of COAD in the CCPs overlapped. Almost 42.13% (525 of 1246) paths of BRCA and 43.03% (525 of 1220) paths of COAD enriched by the CCPs overlapped. Additionally, from the research of Li. et al. [37], we obtained some pathways related to BRCA and COAD, which helped us to understand the similarity between BRCA and COAD. For example, CCP “NUPL2→HDAC4→RUNX3→LEF1→CCND1→E2F2→EZH2→CDKN2A→CDK6→TFDP2→CDT1” in BRCA and CCP “NEK9→NUP205→HDAC4→RUNX3→LEF1→CCND1→E2F2→EZH2→H2AFV” in COAD. Both paths had common parts and specific parts, and they were enriched for breast cancer pathways and colorectal cancer pathways related to BRCA and COAD, respectively, which was reported previously [37]. They were also enriched for pathways in cancer, which was a common core pathway related to the carcinogenesis of various cancers. Additionally, some pathways existed that were related to both BRCA and COAD, such as DNA replication, cell cycle, and p53 signaling pathway. Both TIMELESS→CHEK1→WEE1→CCNE2→CDKN1B→MCM4→POLE→XRCC3 in BRCA and TIPIN→CHEK1→WEE1→CCNE2 in COAD were enriched for the cell cycle pathway and p53 signaling pathway. These indicated that the CCPs we detected had some consistency with those in previous studies. The similarity and high overlapping of the CCPs between BRCA and COAD further demonstrated that control paths detected by CoPath were conserved; likewise, the CCPs in both cancers were enriched for common pathways and specific pathways related to cancers, which indicated that CCPs are effective for reflecting the common and specific features of cancers.

Furthermore, we considered whether drug repositioning with CCPs could provide some insights into a new therapeutic relationship between drugs that already exist and the cancers that we study. We selected five cancers (BRCA, COAD, LIHC, LUAD, and STAD, denoted by BCLLS) with high incidences and mortalities [38], which we prepared to use for drug repositioning. Interestingly, BCLLS also manifested high CCP-based similarity. We believed that signals from drug targets located upstream of CCPs will have an influence downstream of CCPs to achieve a therapeutic effect. We considered three conditions for drug repositioning with CCPs based on drug targets (see Appendix A). Consequently, we acquired 99 drug-cancer pairs consisting of 25 drugs using drug repositioning, of which there were 33 drug-cancer pairs that were FDA-approved and 25 drug-cancer pairs that were implied in previous studies (Appendix A). Then, we used disease genes (from GWAS and OMIM data), C3 module [39], and DIAMOnD module [40] generated from disease genes to perform drug repositioning for BCLLS. Furthermore, we compared the results of drug repositioning obtained for different cases. The condition of drug repositioning using disease genes and the C3 module was more relaxed than CCPs (see Appendix A). The F1-measure of drug repositioning based on CCPs was larger than the results obtained based on disease genes and disease modules for BCLLS (Figure 4b), which indicated that CCPs were more efficient in drug repositioning. We chose drug repositioning with the CCP-based drug target TYMS and DHFR as an example (Figure 4c). We listed six drugs: capecitabine, fluorouracil, trifluridine, and floxuridine (specifically targets protein TYMS); methotrexate (specifically targets protein DHFR); and pemetrexed (specifically targets both proteins TYMS and DHFR). All the targets of these six drugs were located upstream of the CCP shown in Figure 4c. We predicted that their drug effect was transmitted along this path, and they had the ability to intervene in the process of other cancers, such as BCLLS. We also noticed that there was a drug similar to Gemcitabine (an approved drug for BRCA and LUAD), which also targeted TYMS. However, Gemcitabine broadly targeted three proteins. It was not meant to control this path. Therefore, Gemcitabine will not be at the top of the candidate list for BCLLS. For such a scenario, we prioritized drugs whose targets were included in the example CCP in Figure 4c, which showed common drug targets in one CCP of BCLLS, and we predicted a new therapy relationship between cancers and drugs from the CCP, which showed the role of CCPs in drug repositioning.

Additionally, we compared drug repositioning with the proximity method for the C3 module obtained from genes in CCPs and disease genes (see Appendix A). In particular, drug repositioning based on the C3 modules from genes in CCPs was much better than from disease genes for COAD, LIHC, and STAD. When we combined the ROC under different parameters, the results were similar (Figure 4d). We believed that the results were affected by the parameters (the number of drugs we used and upstream genes in the CCPs we defined) because the CCPs also contained specific mechanisms for each cancer but not completely conservative mechanisms among all cancers.

## 4. Discussion

The control paths in complex networks that are related to the dynamic process play a vital role in detecting functional pathways, but little is known about how we can capture common control patterns from a multilayer network. In this paper, first, we developed an approach called CoPath to detect CCPs in a directed multilayer network without cross layers to explain the conserved and specific structure associated with the controllability of complex networks. Second, we proposed an index CV to measure the consistency of control paths in multilayer networks. The CV demonstrated its effectiveness in detecting CCPs on synthetic networks. Third, we applied the CCP pattern to a multilayer cancer network. Our results demonstrated that genes in CCPs had significant biological importance and highlighted the role of CCPs in drug repositioning. There are diversified MMSets and each MMSet illustrates a unique set of control paths. However, enumeration of all MMSets in an arbitrary network is in the class of #P problem. To our knowledge, the novel random sampling method need to obtain nln(n) different MMSets to explore the controllability instead of enumerating all MMSets. Therefore, the time complexity to obtain CCPs is $O\left(n\ln n\times L\times n^{1/2}\times \left|E\right|\right)$, *L* is the number of layers, and $O\left(n^{1/2}\times \left|E\right|\right)$ is the time complexity to compute one maximum matching. This is computationally prohibitive for large networks. Our CoPath can be implemented in low-order polynomial time $O\left(L\times n^{3}\right)$.

## 5. Conclusions

To conclude, our approach provides a new insight for relevant research into multilayer networks. The results demonstrated that CoPath can detect CCPs in a multilayer network efficiently. Furthermore, the approach presented in this paper has many potential applications in numerous areas. For example, CCPs in a multilayer network that consists of different social relationships may reflect a stable social relationship flow. CCPs in a multilayer network that consists of routers in different layers may be core routing links. Our proposed method also has some limitations. For instance, we did not consider a multilayer network with crossing layers. Moreover, CoPath cannot be applied to an undirected graph. Despite this, if there is no need to take controllability into account, CoPath can be extended to an undirected graph easily. We detected CCPs to study the common structure among different cancers and lead to the effective identification of drug targets. We compared our CCP-based method with other disease genes, C3 module, and DIAMOnD module-based method. Prediction accuracy index F1-measure based on CCPs was larger than other methods. The most important thing is that our CCP-based method can provide the pathways through which drugs act. Superior to other methods, it can be used to identify disease treatment mechanisms in biological network.

## Figures and Tables

**Figure 1 entropy-24-00979-f001:**
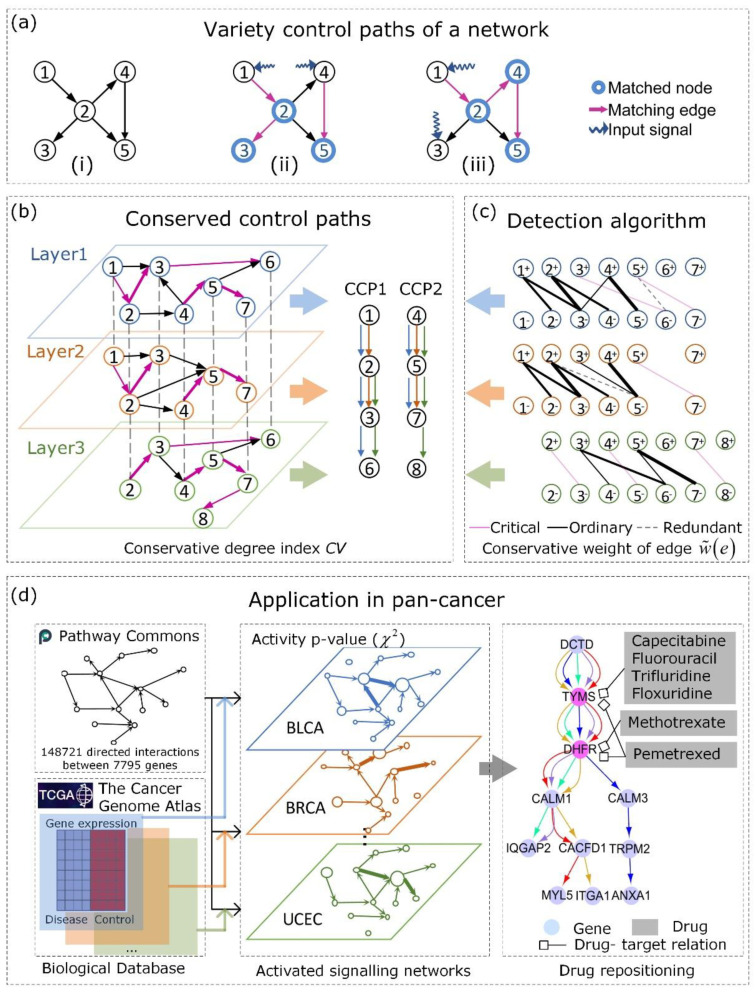
Schematic diagram of CCPs. (**a**) Various control paths of a toy network. (i) A directed network with five nodes and five edges. (ii,iii) The two different maximum matchings for the network in (i). The purple edges that form both MMSets and control paths are 1→2→3, 4→5, and 1→2→4→5. Matched nodes are shown in blue and matching edges are shown in purple. (**b**) The control conserved paths detected by CoPath from a three-layer network. The different colors represent nodes in different layers, and the thickness of the edge in the CCPs represents the weight of the purple edges described by $\sum_{l=1}^{L}I_{e\in M_{l}}\left(e\right)$. Purple edges are matching edges, which form a maximum matching on the corresponding layer, and they can form control paths, which are shown on the right of the figure (**b**). (**c**) The schematic diagram of CoPath. Critical edges, redundant edges, and ordinary edges are represented by a pink solid line, black dashed line, and black solid line, respectively. The bipartite graph of each layer is transformed from the network in the corresponding layer of (**b**). The thickness of ordinary edges reflects the CVs. The more frequently an edge appears in layers that consist of ordinary edges and critical edges, the thicker the edge. (**d**) The application of CCPs in pan-cancer. The far left of (**d**) shows the biological databases we used to construct the network. The middle of (**d**) shows that we constructed a cancer-related activated signaling network with 16 layers. The different colors represent different networks of cancers. The circles are nodes, and the size of the node reflects gene activity in the cancer process. The thickness of the edge reflects edge activity, which is the strength of the regulatory relationship between genes (the definition of edge activity is provided in Appendix A). The far right of (**d**) shows drug repositioning through CCPs, which are detected by CoPath. The purple nodes are drug targets located upstream and the corresponding drug name is surrounded by a gray box.

**Figure 2 entropy-24-00979-f002:**
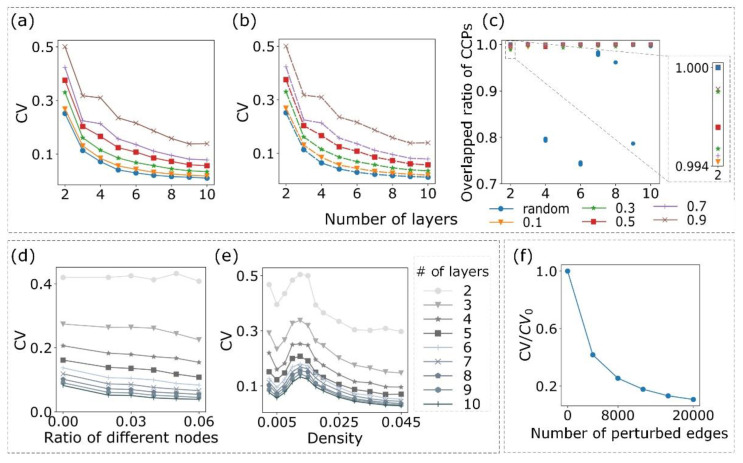
Dependency between the CV and network features. In diagrams (**a**–**c**), the colors and types of lines or points indicate the proportion of common edges between different layers. “Random” in the legend means that the multilayer network is randomly generated without controlling the proportion of common edges. (**a**) Plot of the CV versus the number of layers for random multilayer networks and multilayer networks with different common mechanisms. (**b**,**c**) The same network used in panel (**a**). We consider a random multilayer network or a multilayer network with a given proportion of common edges as a dataset and test it by adding four types of Gaussian noise (μ=0, σ=1, 3, 5, 25) for each network in a dataset. (**b**) Plot of the CV versus the number of layers after adding noise to the edge weight for each edge in the network. The plot is a superposition of the results of four types of noise. (**c**) The ratio of overlapping edges of CCPs before and after adding noise. For a multilayer network with a given number of layers, *L*; we compute the overlapped ratio for the corresponding monoplex network before and after adding noise. There are *L* overlapped ratios, and their median is shown in the plot. Because there are six datasets for a multilayer network with given number of layers *L* and we add four types of noise, there are 24 overlapped ratios for a multilayer with given number of layers *L*. (**d**,**e**) The colors of the lines represent the number of layers of a multilayer network. (**d**) Plot of the CV versus the ratio of difference nodes between two layers. (**e**) Plot of the CV versus the density of the network. The CV reaches the biggest value approximately when the density is around 2e. (**f**) Plot of the CV/CV_0_ ratio of a perturbed multilayer network to a real multilayer network as a function of the number of perturbed edges of the real network. CV_0_ is the CV of the real multilayer network. When the number of crossed edges is zero, which represents the real multilayer network, CV/CV_0_ is 1.

**Figure 3 entropy-24-00979-f003:**
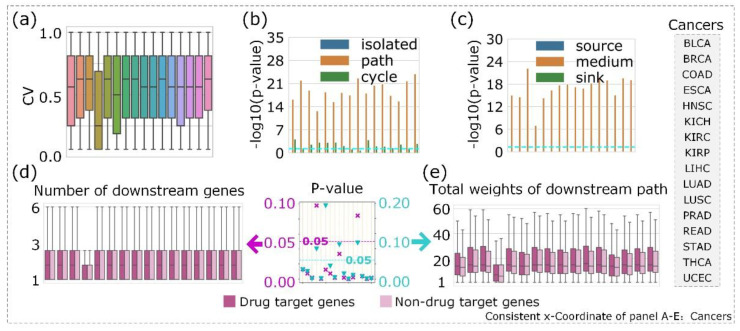
Significant biological importance of genes in the CCPs. In diagrams (**a**–**e**), all plots have a consistent *x*-coordinate which, from left to right, corresponds to the cancer sequence. (**a**) The CV of edges in the CCPs. (**b**) Enrichment analysis of drug target genes based on three positions of genes in the network. The horizontal cyan line represents the value log10(0.05). The *p*-value is derived using the hypergeometric distribution. “Isolated,” “path,” and “cycle” in the figure represent genes that do not occur in the CCPs, genes that occur in the paths of the CCPs, and genes that occur in circular CCPs, respectively. (**c**) Enrichment analysis of drug target genes based on three positions of genes in the CCPs. “Source,” “middle,” and “sink” represent genes that occur at the beginning, middle, and end of the paths of CCPs, respectively. (**d**) Boxplot of the number of downstream genes in CCPs for drug target and non-drug target genes. We do not consider sink nodes in the CCPs because the number of genes downstream of them is zero. (**e**) Boxplot of the conservative value ($\sum_{l=1}^{L}I_{e\in M_{l}}\left(e\right)$) of the downstream path from drug target and non-drug target genes. We calculate statistical significance using the rank-sum test for (**d**,**e**), which is shown in the middle of (**d**,**e**). The magenta cross represents the *p*-value for (**d**) and 87.5% of cancers have *p*-values < 0.05. The cyan inverted triangle represents the *p*-value for (**e**) and 75% of cancers have *p*-values < 0.05. The corresponding horizontal line represents *p*-value = 0.05.

**Figure 4 entropy-24-00979-f004:**
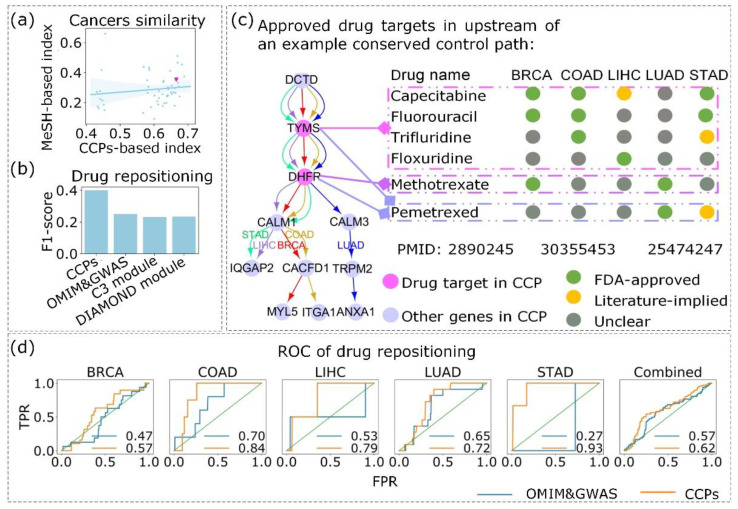
Similarity measurement and the effect of the CCPs in drug repositioning. (**a**) The relationship between MeSH-based similarity and CCP-based similarity. The points represent the similarity between cancers measured using different methods. The line represents the fitting of points, and the shadow area represents the 95% confidence interval. The magenta inverted triangle represents the similarity between BRCA and COAD. (**b**) F1-score of drug repositioning by CCPs, disease genes, and disease modules. (**c**) An example of drug repositioning with CCP-based targets TYMS and DHFR. The CCPs containing targets TYMS and DHFR in BCLLS are shown on the left, and the different colors of edges indicate the edges in the CCPs of different cancers. Circles in different colors shown on the right (green, orange, and gray) represent FDA-approved, literature-implied, and an unclear relationship between the drug and cancer, respectively. We list the PUBMED ID for the three orange circles. The PUBMED IDs from left to right correspond to the three orange circles from top to bottom, in sequence. Areas enclosed with purple, mauve, and blue rectangles represent drug repositioning based on target TYMS, DHFR, and TYMS and DHFR, respectively (e.g., pemetrexed contains targets TYMS and DHFR). (**d**) ROC curves and AUC for BCLLS for different parameters, BRCA (former 30% of genes in the CCP of BRCA and drugs with a z-score < 0), COAD (10%, z-score < 0), LIHC (50%, unlimited z-score), LUAD (10%, z-score < 0), and STAD (50%, z-score < 0). Combined ROC of all 10 ROC curves of BCLLS for an unlimited z-score and z-score < 0, and former 10% of genes in the CCPs of each cancer in BCLLS.

**Table 1 entropy-24-00979-t001:** Scale of the cancer multilayer network classified by edges.

Cancer Type	# Nodes	# Critical Edges	# Redundant Edges	# Ordinary Edges/Nodes
BLCA	5001	333	12,190	32,467/4704
BRCA	5700	255	19,731	53,541/5481
COAD	5528	264	17,337	46,915/5307
ESCA	4295	437	7480	16,603/3830
HNSC	5431	262	16,115	43,560/5209
KICH	5368	326	14,996	39,849/5091
KIRC	5724	254	17,571	54,308/5520
KIRP	5440	287	16,017	43,072/5190
LIHC	5354	250	14,702	41,302/5153
LUAD	5710	262	18,377	49,641/5488
LUSC	5744	263	18,964	51,889/5521
PRAD	5559	273	16,038	50,544/5330
READ	4982	363	11,952	30,957/4632
STAD	5407	335	13,585	39,380/5095
THCA	5561	280	17,389	46,640/5327
UCEC	5538	257	17,202	46,959/5323

# The number of elements in a collection.

**Table 2 entropy-24-00979-t002:** Enrichment analysis of nine functional gene sets.

Functional Gene Set	# Genes	Source
CGC cancer	572	https://cancer.sanger.ac.uk/cosmic/ (accessed on 16 July 2018)
GWAS disease	19,110	http://www.ebi.ac.uk/gwas/ (accessed on 2 January 2018)
OMIM disease	9915	https://omim.org/ (accessed on 8 February 2018)
Virus host	947	http://interactome.dfci.harvard.edu/V_hostome (accessed on 1 January 2018)
Promoter	6222	Kim, T. H. et al. 2005 [33]
Essential	8253	http://tubic.tju.edu.cn/deg/ (accessed on 6 December 2017)
Kinase	516	http://kinase.com/human/kinome (accessed on 31 December 2017)
Drug target	2994	http://www.dgidb.org/ (accessed on 8 January 2018)
Oncogene	119	https://www.oncokb.org/ (accessed on 8 July 2019)

# The number of elements in a collection.

**Table 3 entropy-24-00979-t003:** Most frequently enriched pathways of CCPs.

BLCA		BRCA	
Names of Enriched Pathway	# CCPs	Names of Enriched Pathway	# CCPs
Metabolism	23	Metabolism	33
Metabolism of lipids	14	Metabolism of lipids	19
Phosphatidylinositol signaling system	11	Phosphatidylinositol signaling system	9
Inositol phosphate metabolism	9	T cell receptor signaling pathway	10
Phospholipid metabolism	9	Inositol phosphate metabolism	9
Human T-cell leukemia virus 1 infection	9	Human T-cell leukemia virus 1 infection	9
Glycerophospholipid metabolism	8	Ras signaling pathway	9
T cell receptor signaling pathway	8	PI3K-Akt signaling pathway	9
PI3K-Akt signaling pathway	7	VEGFA-VEGFR2 signaling pathway	9

# The number of elements in a collection.

## Data Availability

The source code of CoPath and all supporting datasets can be downloaded from https://github.com/wangbingbo2019/CCP, including: the 16-layer cancer-related activated signaling network (Appendix A); CCPs of the 16-layer cancer-related activated signaling network (Appendix A); nine functional gene sets (Appendix A); the results of enrichment analysis of genes in the cancer network on nine functional gene sets (Appendix A); drug-cancer pairs predicted using CCPs in BCLLS (Appendix A).

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
