# Peer review of "Conserved Control Path in Multilayer Networks"

_entropy, 2022, doi:10.3390/e24070979_

Round 1

Reviewer 1 Report

This paper proposes a practical conserved control Path detection method, which is built on maximum-cardinality matching with maximum weight algorithm, to discover the paths that play the most consistent roles in controlling signal transmission in multilayer networks. The paper is well written and organized, and the main results seem sound and interesting especially on a multilayered pan-cancer network. This paper makes certain theoretial results, following are some minor concerns that can be considered by the authors:

1)     The authors are enouraged to illustrate the effectiveness and the complexity of the detection algorithm proposed in this paper.

2)     By removing the redundant edges and determining the critical edges, the authors said that the algorithm in this manuscript reduces the running time and improve efficiency, will the removal of redundant edges affect the controllability of the network?

3)     How can the authors guarantee that the output calculated by the Algorithm is the control path?

4)     For complex networks, by what means does the author achieve the classification of edges?

Reviewer 2 Report

1- The structure of the paper is too bad and needs more effort to restructure the paper in a good way.

2- It is better to enhance the abstract to make it more focused.

3- However, the paper is designed to present a practical Conserved control path detection technique which is based on maximum cardinality matching and maximum weigh algorithm, nothing refer to the use of cardinality matching

4-Authors are asked to cite any previous work such as equations, definitions, datasets, ... etc.

5- Authors have to add a conclusion.

6- Update the list of references to include the latest 5 year research results.

7- the methodology isn't clear.

8- Add a description for each data set and justify why you used them.

9- authors are recommended to briefly list and describe the test scenarios, performance metrics/measures, and the performance evaluation.

10- Finally, results need to be validated and compared with others' results.
